# Prospective observational study to examine health-related quality of life and develop models to predict long-term patient-reported outcomes 6 months after hospital discharge with blunt thoracic injuries

Edward Baker ![ORCID],[1,2] Ceri Battle,[3] Abhishek Banjeri,[4] Edward Carlton,[5] Christine Dixon,[6] Jennifer Ferry,[7] Philip Hopkins,[8] Robert Jones,[9] Trevor Murrells,[1] Christine Norton,[1] Lee Patient,[10] Ashraf Rasheed,[11] Imogen Skene ![ORCID],[12] Andrew Tabner ![ORCID],[13] Malcolm Tunnicliff,[14] Louise Young,[15] Andreas Xyrichis ![ORCID],[1] Gerry Lee[1]

For numbered affiliations see end of article.

**Correspondence to**
Mr Edward Baker;
edward.e.baker@kcl.ac.uk

## ABSTRACT

**Objective** This study aimed to examine the long-term outcomes and health-related quality of life in patients with blunt thoracic injuries over 6 months from hospital discharge and develop models to predict long-term patient-reported outcomes.

**Design** A prospective observational study using longitudinal survey design.

**Setting** The study recruitment was undertaken at 12 UK hospitals which represented diverse geographical locations and covered urban, suburban and rural areas across England and Wales.

**Participants** 337 patients admitted to hospital with blunt thoracic injuries were recruited between June 2018–October 2020.

**Methods** Participants completed a bank of two quality of life surveys (Short Form-12 (SF-12) and EuroQol 5-Dimensions 5-Levels) and two pain questionnaires (Brief Pain Inventory and painDETECT Questionnaire) at four time points over the first 6 months after discharge from hospital. A total of 211 (63%) participants completed the outcomes data at 6 months after hospital discharge.

**Outcomes measures** Three outcomes were measured using pre-existing and validated patient-reported outcome measures. Outcomes included: Poor physical function (SF-12 Physical Component Score); chronic pain (Brief Pain Inventory Pain Severity Score); and neuropathic pain (painDETECT Questionnaire).

**Results** Despite a trend towards improving physical functional and pain at 6 months, outcomes did not return to participants perceived baseline level of function. At 6 months after hospital discharge, 37% (n=77) of participants reported poor physical function; 36.5% (n=77) reported a chronic pain state; and 22% (n=47) reported pain with a neuropathic component. Predictive models were developed for each outcome highlighting important data collection requirements for predicting long-term outcomes in this population. Model diagnostics including

## Strengths and limitations of this study

► This study highlights the physical and pain sequelae associated with recovery after blunt thoracic injuries and provides insights into the recovery trajectory for this population.

► The study highlights the complexity of developing prognostic models for long-term outcomes after blunt thoracic injuries and acknowledges the limited current clinical utility in predictive models presented in this study.

► The longitudinal study design using patient-reported outcome measures (PROMs) demonstrates the importance of eliciting health data from this population over throughout the post discharge recovery period.

► Missingness within follow-up data set is a limitation to the prognostic modelling process.

► Despite the benefits of PROMs, it is important to acknowledge the potential inherent bias within these measures.

calibration and discrimination statistics suggested good model fit in this development cohort.

**Conclusions** This study identified the recovery trajectories for patients with blunt thoracic injuries over the first 6 months after hospital discharge and present prognostic models for three important outcomes which after external validation could be used as clinical risk stratification scores.

## BACKGROUND

Blunt thoracic injury (BTI) is a term that covers a spectrum of different injury patterns and severities across all adult age groups.[1] It accounts for about 15% of injured patients

presenting to trauma hospitals, with mortality rates reported between 4%–20%.[2] Outcomes within trauma care have historically been measured primarily using mortality as a key outcome.[3] During the last two decades, there has been a decrease in the mortality rates of patients with polytrauma in high-income countries with organised trauma systems.[4] As the mortality rates have decreased, there has been a growing focus on measuring outcomes relating to injury morbidity.[3 5] There is a growing body of international research investigating the long-term outcomes of patients with BTI, but within the UK, there is a paucity of BTI outcomes research with little consistency in the methods employed.[6 7]

For patients recovering from BTI, it has been identified that reduced physical function, persistent altered respiratory function and chronic pain frequently impact on all aspects of daily living.[6] Health-related quality of life (HRQoL) provides a strong foundation for measuring the impact of these symptoms with a holistic and comprehensive approach.[8] It is important not to isolate individual components of daily living but consider them as a single entity, and HRQoL measurement provides a vehicle to do this.[9] Furthermore, it allows the development of theory driven interventions that impact on all elements of daily living.[10] For the purposes of this study HRQoL is defined as including all aspects of self-perceived well-being that are related to, or affected by, the presence of disease or treatment.[11]

Follow-up after hospital discharge with major trauma is often limited within the UK's National Health Service (NHS), resulting in limited clinical knowledge about the recovery journey of patients with BTI.[12] This also leaves clinicians without a trajectory for recovery making it very difficult to adequately educate and prepare patients for self-management in the post hospital discharge recovery period.[13] Evidence from this study will be used to facilitate the identification of patients at risk of poor outcomes after BTI, help develop health promotion materials and to support the development of follow-up interventions within BTI management.[13]

Having the ability to predict outcomes and develop methods for risk stratification is extremely important in trauma care.[14] There is a growing body of prognostic research in this area with simple scoring systems developed and validated with effective predictive abilities across different health systems.[2] Where resources for patient follow-up are limited, it is key that effective methods for identifying patients in high-risk groups for complications or poor outcomes are in place, allowing resources to be focused on these patients at the greatest risk. The aim of this study was to examine the long-term outcomes and HRQoL in patients with BTI over 6 months from hospital discharge and develop models to predict long-term patient-reported outcomes. Study objectives were as follows:

► Investigate changes in HRQoL over 6 months after hospital discharge with BTI.
► Examine the evolution of pain and neuropathic pain over 6 months after discharge from hospital with BTI.
► Develop predictive models for poor physical functioning, chronic pain and neuropathic pain at 6 months after hospital discharge with BTI.

## METHODS

This prospective multicentred observational study employed longitudinal survey methods to follow-up patients recovering from BTI over a 6-month period from hospital discharge. This research forms the quantitative component of a mixed methods study investigating recovery after BTI. The qualitative components will be reported separately.[15]

### Study setting

The study population sample was recruited from major trauma centres (Level 1) (n=5) and trauma units (Level 2–3) (n=7) from diverse geographical locations within urban, suburban and rural areas across England and Wales. Sites were set up in a stepwise fashion and in total 12 acute hospital sites recruited participants between 6–10 months in two separate recruitment cohorts between June 2018 and March 2020.

### Eligibility criteria
#### Inclusion criteria
► Participant is willing and able to give informed consent for participation in the study
► Male or female, aged 16 years or above
► Admitted to a major trauma centre/trauma unit with primary BTI.
► Admitted to hospital for a period of 24 hours or greater.

#### Exclusion criteria
► Unstable spinal fracture or spinal cord injury.
► Traumatic brain injury with cognitive impairment
► Altered mental status that would prevent accurate self-assessment through questionnaires or prevent informed consent into study
► Associated unstable pelvic fracture or open abdomen requiring extended periods of treatment in hospital
► Limb injuries from same traumatic event that require surgical intervention
► Participant is a prisoner/in police custody at the time of recruitment.

### Data collection
Data collection was undertaken at four time points: (1) Hospital discharge (including a pre-injury estimation of HRQoL); (2) 1 month after discharge; (3) 3 months after discharge; and (4) 6 months after discharge. Potential participants were identified by the local clinical team or the clinical research nurses at each site. Demographic and injury-related data were collated throughout the acute admission using a previously tested case report form,[1] and site staff conducted data collection at hospital discharge. Post discharge follow-up was conducted by the central research team at the host university using either postal surveys or electronic surveys hosted by SurveyMonkey.

During post discharge follow-up non-responding participants were contacted up to three times using a mixed multimedia approach including telephone, post or email, to maximise participation and minimise the potential for missing data.

## Ethics considerations

Formal informed written consent was taken from all participants after information sheets had been provided and sufficient time was allowed for discussion with family and clinical research staff. Participants were informed that everyone who completed data collection at all time points would be placed in a prize draw to win a £50 shopping voucher to recognise the time associated with participation in this longitudinal follow-up study.

## Sample size

This observational study had an exploratory design and pragmatic approach to sample size. Although there is no consensus on the sample requirements for logistic regression analysis, based on previous studies we aimed to have 10 events for each predictor variables included in the model.[2 3] As it was likely that 10 predictor variables would be used in the modelling process for each model, the aim was to achieve 100 events (ie, outcomes of interest) in the sample.

## Outcome measures

HRQoL was explored first at a global level through the Physical Component Score (PCS) and Mental Component Score (MCS) of the Short Form-12 (SF-12),[16] and second, using the individual levels of the EuroQol 5-Dimensions 5-Levels (EQ-5D-5L).[17 18] Pain was previously recognised as an ongoing problem throughout post hospital discharge recovery.[6] In this study, pain severity and impact were measured using the Brief Pain Inventory (BPI)[19] and neuropathic pain was assessed using the pain-DETECT Score[20] Using existing literature on HRQoL after BTI,[6 21] three key outcomes were selected:

1. **Physical Function** is an individual's ability to undertake actions that involve physical activities, ranging from activities of daily living to more complex activities that involve a combination of skills, often within a social context.[3] For the purposes of this study, physical functioning was measured using the PCS of the SF-12 at 6 months after discharge from hospital. A PCS of ≤35 (range: 0–100 with higher scores equalling better physical function) was taken as defining poor physical functioning from previous literature.[16]

2. **Chronic Pain** is defined as pain that persists beyond the normal expected healing time and therefore lacks the acute warning function of physiological nociception. Pain is normally considered as chronic when it lasts or recurs for more than 3–6 months.[22] For this study, chronic pain was defined as having a BPI Pain Severity Score (PSS) of ≥3.5 (range: 0–10 and higher scores equals increased pain severity) at 6 months after hospital discharge.[23]

3. **Neuropathic pain** is defined as pain caused by a lesion or disease of the somatosensory nervous system.[24] For the purposes of this study, participants with a potential neuropathic pain component were identified using the painDETECT Questionnaire (PDQ).[20] A painDETECT score ≥12 (range: −1 to 38) at 6 months after hospital discharge defined our cohort of patients with a potential neuropathic pain component.[25]

Although the limitations of using binary outcome measures were acknowledged in the planning phase, this was an important component in the process of developing prognostic models which could be used as risk stratification scores in the clinical practice setting.

## Selection of candidate predictor variables

A collection of predictor variables was required which covered multiple potentially influential factors relating to the outcomes of interest. Initially, during the study development, a long list of candidate variables were selected from previous literature.[6 10 26 27] From this long-list, shortlists of variables for each outcome were compiled. Decision-making for entering variables into the modelling process was based on their clinical significance and achieving statistical significance at the 5% level in the univariate analysis phase. There was an 'a priori' plan for candidate variables to be selected in three phases, initially using only variables that are readily available in clinical practice, then using variables that are currently not readily available in clinical practice and finally using variables collected at post discharge time points.

## Statistical analysis

Data analysis was undertaken using SPSS V.26. Statistical data were presented using descriptive and inferential statistical methods and reported according to the Strengthening the Reporting of Observational Studies in Epidemiology statement.[28] Binary and categorical data were presented as percentages and number of cases. Continuous data were presented as means with SD. The statistical tests applied for binary/categorical data were $\chi^2$ test of association, except where the expected cell counts were ≤5 when the Fisher's exact test was used. Comparisons continuous data from two independent groups was undertaken using the Student's t-test. Longitudinal changes in continuous variables over the course of the post discharge follow-up period are reported at a group level. This uses the longitudinal change in the group mean from one time point to the next to measure change.

Missing data were addressed using the multiple imputation procedure in SPSS. Most missing data were found within the follow-up phase of data collection where units of data (ie, surveys) were not returned. The imputation model included all predictor variables with potential for inclusion in the modelling process including candidate predictor variables from within the hospital admission and from the post discharge follow-up. A total of 10 imputed data sets were created. Longitudinal imputation of outcome variables was considered but not conducted

in this study due to the high level of missingness at both 1 month and 3 months and the low number of participants who completed the follow-up data collection at all time points. Ultimately despite the potential limitations, a case wise approach was used.

The predictive model framework was developed in accordance with the TRIPOD (Transparent Reporting of multivariable predication model for Individual Prognosis Or Diagnosis) methodology[29] and the study methods based on previously published prognostic modelling studies.[2 3 30] A multivariable logistic regression analysis was used to evaluate the association of the predictors with the outcome (dependent variable) in each imputed data set using the forward stepwise method.[31] In this process, SPSS automatically produced a final model that aggregated the results across all the imputed data sets.

Calibration was evaluated using a Hosmer-Lemeshow test where a p value>0.05 was considered to represent good calibration. In addition, graphical calibration plots were developed. A receiver operating characteristic (ROC) curve was plotted for each model. A c-statistic (area under the receiver operator curve) was calculated, that tested whether the amount of discrimination was significantly different from random (c-statistic of 0.5 indicates that discrimination is no better than the toss of a coin). Performance of the final model was assessed using sensitivity and specificity.[32] Throughout statistical significance was recognised at the 5% level (p<0.05).

### Patient and public involvement

A patient and public reference group was formed and involved in the development phases of this study including during protocol development and application for ethical approval but have not been involved in data collection, analysis, or interpretation.

### RESULTS

A total of 337 patients were recruited to the study. The mean age of participants was 62 years (±16.5) and 69% (n=231) were men. The predominant mechanism of injury was falls <2 m (45%, n=153) and the mean number of rib fractures was 5.2 (±4). At 6 months after discharge, 37% (n=77) of the sample self-reported poor physical functioning, 37% (n=77) reported chronic pain and 22% (n=47) reported pain with a likely neuropathic pain component. Table 1 summarises key demographic data for the total sample. Further greater detailed demographic data are presented in online supplemental table S1 as a supplementary file to this study.

### Missing data analysis

Overall, the combined response rate at all time points was 72%. At hospital discharge 95% (n=320) completed the discharge data collection time point, subsequently, 62% (n=210) and 43% (n=144) completed data collection at 1 month and 3 months after discharge, respectively. At 6 months, 63% (211) of the total sample completed

follow-up where the study outcomes data were collected. Of the 211 participants who completed the 6-month follow-up, n=200 (94%) completed discharge data collection, 174 (82%) completed the 1 month data collection time point and 142 (67%) completed data collection at 3 months after discharge.

Overall missingness within the data set was predominately from 'unit missing data' (ie, completed survey not returned by participant) as individual items of missing data within a returned survey were followed up with participants at the time of data collection. Among candidate predictor variables 17% (n=37) were missing at 1 month follow-up, and 33% (n=69) at 3 months. Although this represents high levels of missingness within the data set, this is not uncommon in longitudinal follow-up studies after traumatic injuries .[6]

A missing data analysis was undertaken as part of the analysis process to identify any differences in the characteristics of participants who did or did not respond to follow-up at 6 months after hospital discharge when outcome data were collected. Online supplemental table S2 presents the results from this analysis. Although there was a statistically significant difference in participants age, incidence of cardiac disease and cancer and MCS at the pre-injury, discharge and 1 month post discharge time point, these were not deemed to be clinically significant differences. Furthermore, a logistic regression model was built using a stepwise forward approach using variables identified in the univariable analysis of missingness to identify predictors of responder status at 6 months after discharge. It was not possible to fit a model using the selected variables and no statistically significant predictors were identified. Since none were identified, it was assumed that the two groups were characteristically similar and therefore a sensitivity analysis, using weights calculated from the logistic regression model, was not undertaken. Based on Little and Rubin's taxonomy missingness was assumed to be missing at random (MAR) at the unit level and multiple imputation methods in SPSS were used to address item level missingness .[33 34]

### Changes in HRQoL over 6 months after discharge from hospital

Physical function was clearly affected by the injuries experienced by this cohort. Measures of HRQoL are presented in table 2. As expected, the mean PCS score (SF-12) saw a 19.3% (46.5–27.2) drop from the pre-injury estimate through to the initial hospital discharge time point representing a substantial negative change in physical functioning. Over the 6-month follow-up there were incremental improvements in the mean PCS score (31.6, 40.1 and 41.7 at 1, 3 and 6 months, respectively) but the sample failed to return to their baseline estimate of physical function (−4.8%) or population norms (−8.3%) within the 6-month follow-up period. During the same time frame, the impact of injury on mental health was lesser than on physical function. The mean MCS score (SF-12) dropped from 49.7 to 44.4 (−5.3) from the

**Table 1** Demographics of entire cohort and a comparison of characteristics for each outcome of interest at 6 months after discharge

| Result: n= (%) unless otherwise indicated | Total sample (n=337) | Poor physical function (n=77) | Good physical function (n=134) | P value | Chronic pain (n=77) | No chronic pain (n=134) | P value | Neuropathic component likely (n=47) | Neuropathic component unlikely (n=164) | P value |
|---|---|---|---|---|---|---|---|---|---|---|
| Age (mean (SD)) | 62 (16.5) | 69.1 (13.9) | 60.8 (16.3) | <0.001 | 66.7 (16.4) | 62.1 (15.6) | 0.05 | 63.4 (17) | 63.9 (15.7) | 0.84 |
| Male (%) | 231 (68.5) | 49 (63.6) | 97 (72.4) | 0.19 | 54 (70.1) | 91 (67.9) | 0.80 | 33 (70.2) | 112 (68.7) | 0.85 |
| Injury severity score (mean (SD)) | 11.2 (7.3) | 10.9 (7.8) | 11.4 (6.8) | 0.61 | 11.3 (7.9) | 11.2 (6.8) | 0.89 | 11.6 (8.3) | 11.2 (6.9) | 0.76 |
| **Mechanism of injury:** | | | | | | | | | | |
| RTC | 78 (23.1) | 14 (18.2) | 38 (28.4) | 0.23 | 19 (24.7) | 33 (24.6) | 0.78 | 9 (19.1) | 43 (26.4) | 0.82 |
| Crush | 11 (3.3) | 3 (3.9) | 4 (3) | | 1 (1.3) | 6 (4.5) | | 1 (2.1) | 6 (3.7) | |
| Ped versus vehicle | 4 (1.2) | 1 (1.3) | 2 (1.5) | | 1 (1.3) | 2 (1.5) | | 1 (2.1) | 2 (1.2) | |
| Fall >2 m | 69 (20.5) | 13 (16.9) | 30 (22.4) | | 18 (23.4) | 25 (18.7) | | 13 (27.7) | 30 (18.4) | |
| Fall <2 m | 153 (45.4) | 44 (57.1) | 54 (40.3) | | 37 (48.1) | 61 (45.5) | | 22 (46.8) | 76 (46.6) | |
| Assault | 14 (4.2) | 1 (1.3) | 4 (3) | | 1 (1.3) | 3 (2.2) | | 1 (2.1) | 3 (1.8) | |
| CPR | 4 (1.2) | 1 (1.3) | 0 | | 0 | 1 (0.7) | | 0 | 1 (0.6) | |
| Rapid deceleration | 4 (1.2) | 0 | 2 (1.5) | | 0 | 2 (1.5) | | 0 | 2 (1.2) | |
| **Comorbidities:** | | | | | | | | | | |
| Respiratory comorbidity | 70 (20.8) | 21 (27.3) | 21 (15.7) | 0.042 | 21 (27.3) | 20 (14.9) | 0.031 | 12 (25.5) | 29 (17.8) | 0.24 |
| Cardiac comorbidity | 147 (43.6) | 49 (63.6) | 53 (39.6) | 0.001 | 47 (61) | 54 (40.3) | 0.004 | 27 (57.4) | 74 (45.4) | 0.15 |
| Neuro comorbidity | 50 (14.8) | 20 (26) | 9 (6.7) | <0.001 | 19 (24.7) | 10 (7.5) | 0.001 | 10 (21.3) | 19 (11.7) | 0.092 |
| MSK comorbidity | 89 (26.4) | 26 (33.8) | 35 (26.1) | 0.24 | 30 (39) | 31 (23.1) | 0.016 | 16 (34) | 45 (27.6) | 0.39 |
| Pre-injury regular analgesic use | 61 (18.1) | 23 (29.9) | 15 (11.2) | 0.001 | 30 (39) | 8 (6) | <0.001 | 19 (40.4) | 19 (11.7) | <0.001 |
| **Injury descriptors:** | | | | | | | | | | |
| No. rib fractures (mean (SD)) | 5.2 (4) | 5.3 (3.8) | 5.3 (4) | 0.90 | 5.4 (3.9) | 5.3 (3.9) | 0.74 | 5.6 (4.1) | 5.2 (3.8) | 0.55 |
| Operative rib fixation | 16 (4.7) | 6 (7.8) | 3 (2.2) | 0.05 | 2 (2.6) | 7 (5.2) | 0.36 | 3 (6.4) | 6 (3.7) | 0.42 |
| First rib fracture | 31 (9.2) | 6 (7.8) | 13 (9.7) | 0.64 | 10 (13) | 9 (6.7) | 0.13 | 7 (14.9) | 12 (7.4) | 0.11 |
| Bilateral rib fractures | 44 (13.1) | 9 (11.7) | 19 (14.2) | 0.61 | 11 (14.3) | 17 (12.7) | 0.76 | 10 (21.3) | 18 (11) | 0.07 |
| Radiological flail | 60 (17.8) | 13 (16.9) | 29 (21.6) | 0.41 | 16 (20.8) | 26 (19.4) | 0.83 | 7 (14.9) | 35 (21.5) | 0.32 |
| Extra-thoracic injuries | 104 (30.9) | 25 (32.5) | 35 (26.1) | 0.33 | 19 (24.7) | 40 (29.9) | 0.40 | 12 (25.5) | 47 (28.8) | 0.66 |
| **Analgesic mode:** | | | | | | | | | | |
| Intermittent analgesia only | 179 (53.1) | 37 (48.1) | 77 (57.5) | 0.19 | 37 (48.1) | 76 (56.7) | 0.20 | 22 (46.8) | 91 (55.8) | 0.28 |
| Regional analgesia | 106 (31.5) | 29 (37.7) | 41 (30.6) | 0.29 | 31 (40.3) | 39 (29.1) | 0.11 | 20 (42.6) | 50 (30.7) | 0.13 |
| Patient-controlled analgesic only | 93 (27.6) | 23 (29.9) | 29 (21.6) | 0.18 | 20 (26) | 32 (23.9) | 0.76 | 12 (25.5) | 40 (24.5) | 0.89 |
| **Short-term inpatient outcomes:** | | | | | | | | | | |

Continued

**Table 1** Continued

| Result: n= (%) unless otherwise indicated | Total sample (n=337) | Poor physical function (n=77) | Good physical function (n=134) | P value | Chronic pain (n=77) | No chronic pain (n=134) | P value | Neuropathic component likely (n=47) | Neuropathic component unlikely (n=164) | P value |
|---|---|---|---|---|---|---|---|---|---|---|
| Complications: | 111 (32.9) | 34 (44.2) | 33 (24.6) | 0.003 | 25 (32.5) | 42 (31.3) | 0.89 | 18 (38.3) | 49 (30.1) | 0.29 |
| Post admission ventilation: | 26 (7.7) | 9 (11.7) | 5 (3.7) | 0.025 | 6 (7.8) | 8 (6) | 0.62 | 5 (10.6) | 9 (5.5) | 0.22 |
| ICU length of stay (mean (SD)) | 7.6 (7.4) | 9.42 (7.5) | 6.42 (5.8) | 0.16 | 7.6 (5.4) | 7.8 (7.3) | 0.93 | 9.4 (7.2) | 7.4 (6.6) | 0.49 |
| Hospital length of stay (mean (SD)) | 8.9 (9.5) | 11.7 (10.2) | 6.9 (5.7) | <0.001 | 10.4 (9.3) | 7.7 (7) | 0.03 | 9.9 (7.2) | 8.3 (8.2) | 0.21 |

CPR, cardio-pulmonary resuscitation; ICU, intensive care unit; MSK, musculoskeletal; Ped, pedestrian; RTC, road traffic collision.

pre-injury estimate through to hospital discharge, dropping a further 0.5% to 43.9 at 1 month after hospital discharge. Between the 1 and 6 months the mean MCS increased to 47.8 which remained 1.9% below the pre-injury estimate and 2.2% below the population norms for England.

The impact of BTI on individual components of HRQoL were measured using the EQ-5d-5L questionnaire. Table 3 presents the individual dimension/level data for each time point. At hospital discharge, the lowest scored (best) dimension was anxiety and depression with 43% of the sample reporting no problems in this dimension. At 6 months after discharge: 'Mobility', 'Self-Care', 'Usual Activities' and 'Anxiety/Depression' were the best rated dimensions, with between 42% and 73% of the sample reporting no problems in these areas. The highest scoring (worst) dimensions at hospital discharge were 'Usual Activities' where 42% (n=135) reported being wholly unable to complete usual activities. Between 42%–43% of the sample reported moderate (3), severe (4) or extreme (5) pain at both 3 months and 6 months after discharge from hospital (42.3% (n=61) at 3 months and 43.1% (n=91) at 6 months). At 6 months after injury, there was a proportion of participants who reported severe or extreme problems in all dimensions, and these exceeded the pre-injury estimated levels. This demonstrates that although the 6-month follow-up period was sufficient to observe recovery in a vast proportion of this sample, there were participants who reported extreme disability beyond the follow-up.

### Evolution of chronic pain and pain with a neuropathic component

Subjective pain evaluated through the BPI PSS reflects the persistent levels of pain experienced by study participants throughout the 6-month study follow-up period. Pain-related outcome measures are presented in table 2. Although there were consistent incremental reductions in the mean pain score over time, at 6 months after injury the mean pain severity score was 2.7 (SD 2.7); 36.5% (n=77) of participants were reporting chronic pain. The mean BPI Pain Interference Score (PIS) also demonstrated incremental decreases until the 3-month time point where there appears to be a plateau, followed by a marginal deterioration of the mean PIS (2.6 (SD 2.7) to 2.7 (SD 2.9)). This highlights that participants reporting pain that impacted on daily activities at 3 months were likely to continue to experience these interferences at 6 months after hospital discharge. Similarly, mean self-reported analgesic effectiveness decreased from 65.3 (SD 24.8) at hospital discharge to 46.5 (SD 35.8) at 3 months which then plateaued and was 49.2 (SD 33.4) at 6 months after hospital discharge. Individual components of the BPI PSS and BPI PIS can be seen in the online supplemental tables S3 and S4.

At hospital discharge and 1 month after, 30.6% (n=98) and 32.4% (68%) (respectively) of patients reported moderate or high risk of having a neuropathic pain

**Table 2** Cohort outcomes with comparisons between outcomes of interest at 6 months after discharge

| | Outcome | Total sample (n=337) | Poor physical function (n=77) | Good physical function (n=134) | P value | Chronic pain (n=77) | No chronic pain (n=134) | P value | Neuropathic pain likely (n=47) | Neuropathic pain unlikely (n=164) | P value |
|---|---|---|---|---|---|---|---|---|---|---|---|
| Pre-injury | SF-12 PCS* | 46.5 (11) | 40 (11) | 50.8 (7.7) | <0.001 | 41.9 (11.4) | 49.6 (8.6) | <0.001 | 41.7 (11.5) | 48.3 (9.6) | <0.001 |
| | SF-12 MCS* | 49.7 (10.7) | 48.2 (11.7) | 53.1 (7.6) | <0.001 | 49.1 (10.9) | 52.7 (8.4) | 0.009 | 48.2 (10.6) | 52.3 (9.1) | 0.009 |
| Discharge (n=320) | SF-12 PCS* | 27.2 (8.6) | 24.3 (7.4) | 28.4 (8.6) | 0.001 | 25.1 (8.3) | 27.9 (8.3) | 0.02 | 25.1 (8.4) | 27.5 (8.3) | 0.09 |
| | SF-12 MCS* | 44.4 (11.8) | 43.5 (11.8) | 47 (11.5) | 0.04 | 46.1 (11.5) | 45.5 (11.9) | 0.76 | 42.6 (11.7) | 46.7 (11.6) | 0.04 |
| | BPI pain severity score* | 5 (2.1) | 5.3 (2.1) | 4.8 (2) | 0.10 | 5.3 (2) | 4.7 (2) | 0.06 | 5.6 (2) | 4.7 (2) | 0.009 |
| | BPI pain interference score* | 5.7 (2.3) | 6.5 (2) | 5.2 (2.4) | <0.001 | 5.9 (2) | 5.5 (2.4) | 0.18 | 6.4 (2) | 5.4 (2.3) | 0.004 |
| | PDQ: low risk† | 222 (69.4) | 48 (64.9) | 95 (73.1) | 0.42 | 47 (63.5) | 96 (74.4) | 0.075 | 24 (52.2) | 119 (75.8) | 0.006 |
| | PDQ: moderate risk† | 67 (20.9) | 18 (24.3) | 26 (20) | | 22 (29.7) | 21 (16.3) | | 17 (37) | 26 (16.6) | |
| | PDQ: high risk† | 31 (9.7) | 8 (10.8) | 9 (6.9) | | 5 (6.8) | 12 (9.3) | | 5 (10.9) | 12 (7.6) | |
| 1 month (n=210) | SF-12 PCS* | 31.6 (7.8) | 27 (6) | 33.5 (7.5) | <0.001 | 27.7 (6) | 32.9 (7.8) | <0.001 | 27.8 (5.8) | 32.1 (7.9) | <0.001 |
| | SF-12 MCS* | 43.9 (11.8) | 40.8 (12.2) | 46.8 (11.2) | 0.001 | 4.07 (12.7) | 46.7 (10.9) | 0.001 | 35.6 (12.3) | 47.0 (10.6) | <0.001 |
| | BPI pain severity score* | 3.8 (2.2) | 4.7 (2.3) | 3.2 (1.9) | <0.001 | 5.2 (2) | 3 (1.9) | <0.001 | 5.5 (2.2) | 3.3 (1.9) | <0.001 |
| | BPI pain interference score* | 4.4 (2.7) | 5.6 (2.6) | 3.7 (2.5) | <0.001 | 5.8 (2.4) | 3.6 (2.5) | <0.001 | 5.6 (2.4) | 3.8 (2.4) | <0.001 |
| | PDQ: low risk† | 142 (67.6) | 39 (61.9) | 83 (73.5) | 0.14 | 34 (57.6) | 88 (75.2) | 0.005 | 14 (38.9) | 108 (77.1) | <0.001 |
| | PDQ: moderate risk† | 45 (21.4) | 14 (22.2) | 22 (19.5) | | 13 (22) | 23 (19.7) | | 9 (25) | 27 (19.3) | |
| | PDQ: high risk† | 23 (11) | 10 (15.9) | 8 (7.1) | | 12 (20.3) | 6 (5.1) | | 13 (36.1) | 5 (3.6) | |
| 3 months (n=144) | SF-12 PCS* | 40.1 (11) | 31.4 (8.8) | 44.4 (9.5) | <0.001 | 32.4 (9.1) | 43.9 (10) | <0.001 | 32.9 (8.8) | 42.1 (10.8) | <0.001 |
| | SF-12 MCS* | 48.3 (11.4) | 43.1 (12.9) | 50.9 (9.7) | <0.001 | 43.1 (11.8) | 51 (10.4) | <0.001 | 40 (12) | 50.7 (10.2) | <0.001 |
| | BPI pain severity score* | 2.9 (2.6) | 4.4 (2.7) | 2.2 (2.2) | <0.001 | 5.1 (2.2) | 1.8 (2) | <0.001 | 4.8 (2.2) | 2.4 (2.4) | <0.001 |
| | BPI pain interference score* | 2.6 (2.7) | 4.4 (3.1) | 1.8 (2.1) | <0.001 | 5 (2.6) | 1.5 (2) | <0.001 | 5 (2.6) | 2 (2.4) | <0.001 |
| | PDQ: low risk† | 110 (76.4) | 28 (60.9) | 79 (83.2) | 0.012 | 23 (48.9) | 84 (89.4) | <0.001 | 9 (29) | 98 (89.1) | <0.001 |
| | PDQ: moderate risk† | 15 (10.4) | 7 (15.2) | 8 (8.4) | | 9 (19.1) | 6 (6.4) | | 9 (29) | 6 (5.5) | |
| | PDQ: high risk† | 19 (13.2) | 11 (23.9) | 8 (8.4) | | 15 (31.9) | 4 (4.3) | | 13 (41.9) | 6 (5.5) | |
| 6 months (n=211) | SF-12 PCS* | 41.7 (11.5) | 29.2 (5.1) | 48.9 (7.2) | <0.001 | 32.9 (8) | 46.8 (10.1) | <0.001 | 32.4 (8.5) | 44.4 (10.9) | <0.001 |
| | SF-12 MCS* | 47.8 (11.8) | 42.3 (13.2) | 51 (9.7) | <0.001 | 39.4 (11.6) | 52.7 (8.9) | <0.001 | 36.4 (11.8) | 51.1 (9.6) | <0.001 |
| | BPI pain severity score* | 2.7 (2.7) | 4.8 (2.6) | 1.5 (2) | <0.001 | 5.8 (1.8) | 0.9 (1.1) | <0.001 | 5.4 (2.3) | 1.9 (2.3) | <0.001 |
| | BPI pain interference score* | 2.8 (2.9) | 5.2 (2.7) | 1.4 (2) | <0.001 | 5.6 (2.3) | 1.2 (1.7) | <0.001 | 6.1 (2.6) | 1.8 (2.2) | <0.001 |
| | PDQ: low risk† | 164 (77.7) | 44 (57.9) | 119 (88.8) | <0.001 | 40 (51.9) | 123 (92.5) | <0.001 | 0 | 163 (100) | <0.001 |
| | PDQ: moderate risk† | 28 (13.3) | 17 (22.4) | 11 (8.2) | | 20 (26) | 8 (6) | | 28 (59.6) | 0 | |
| | PDQ: high risk† | 19 (9) | 15 (19.7) | 4 (3) | | 17 (22.1) | 2 (1.5) | | 19 (40.4) | 0 | |

*Mean (±SD).
†n (%).
BPI, Brief Pain Inventory; MCS, Mental Component Score; PCS, Physical Component Score; PDQ, painDETECT Questionnaire; SF-12, Short Form-12 Health Survey.

**Table 3** Individual dimension and level data from EuroQoL 5-Dimensions (5-Levels)

| (n= (%)) | No problems | Slight problems | Moderate problems | Severe problems | Unable/extreme problems |
|---|---|---|---|---|---|
| Mobility: | | | | | |
| Pre-injury | 204 (60.5) | 50 (14.8) | 47 (13.9) | 33 (9.8) | 3 (0.9) |
| Discharge | 50 (15.6) | 78 (24.4) | 102 (31.9) | 72 (22.5) | 18 (5.6) |
| 1 month | 65 (31) | 76 (36.2) | 38 (18.1) | 26 (12.4) | 5 (2.4) |
| 3 months | 71 (49.3) | 36 (25) | 17 (11.8) | 16 (11.1) | 4 (2.8) |
| 6 months | 110 (52.1) | 41 (19.4) | 32 (15.2) | 23 (10.9) | 5 (2.4) |
| Self-care: | | | | | |
| Pre-injury | 269 (79.8) | 33 (9.8) | 23 (6.8) | 10 (3) | 2 (0.6) |
| Discharge | 68 (21.3) | 92 (28.8) | 81 (25.3) | 59 (18.4) | 20 (6.3) |
| 1 month | 108 (51.4) | 59 (28.1) | 30 (14.3) | 10 (4.8) | 3 (1.4) |
| 3 months | 106 (73.6) | 21 (14.6) | 9 (6.3) | 6 (4.2) | 2 (1.4) |
| 6 months | 153 (72.5) | 31 (14.7) | 20 (9.5) | 5 (2.4) | 2 (0.9) |
| Usual activities: | | | | | |
| Pre-injury | 224 (66.5) | 53 (15.7) | 33 (9.8) | 19 (5.6) | 8 (2.4) |
| Discharge | 21 (6.6) | 36 (11.3) | 72 (22.5) | 56 (17.5) | 135 (42.2) |
| 1 month | 20 (9.5) | 95 (45.2) | 41 (19.5) | 29 (13.8) | 25 (11.9) |
| 3 months | 49 (34) | 51 (35.4) | 22 (15.3) | 9 (6.3) | 13 (9) |
| 6 months | 89 (42.2) | 62 (29.4) | 34 (16.1) | 14 (6.6) | 12 (5.7) |
| Pain/discomfort: | | | | | |
| Pre-injury | 157 (46.6) | 94 (27.9) | 61 (18.1) | 18 (5.3) | 7 (2.1) |
| Discharge | 14 (4.4) | 41 (12.8) | 151 (47.2) | 74 (23.1) | 40 (12.5) |
| 1 month | 15 (7.1) | 49 (23.3) | 112 (53.3) | 22 (10.5) | 12 (5.7) |
| 3 months | 37 (25.7) | 46 (31.9) | 47 (32.6) | 10 (6.9) | 4 (2.8) |
| 6 months | 72 (34.1) | 48 (22.7) | 72 (34.1) | 11 (5.2) | 8 (3.8) |
| Anxiety/depression: | | | | | |
| Pre-injury | 188 (55.8) | 79 (23.4) | 54 (16) | 11 (3.3) | 5 (1.5) |
| Discharge | 138 (43.1) | 85 (26.6) | 64 (20) | 20 (6.3) | 13 (4.1) |
| 1 month | 100 (47.6) | 38 (18.1) | 57 (27.1) | 7 (3.3) | 8 (3.8) |
| 3 months | 82 (56.9) | 27 (18.8) | 26 (18.1) | 4 (2.8) | 5 (3.5) |
| 6 months | 119 (56.4) | 42 (19.9) | 33 (19.9) | 8 (3.8) | 9 (4.3) |

component. At 3 months and 6 months, there was a reduction in the number of participants reporting neuropathic pain (23.6% (n=34) and 22.3% (n=47), respectively, with 13.2% (n=19) and 9.0% (n=19), respectively, reporting painDETECT scores in the high-risk category (representing 90% likelihood of having a neuropathic pain component).

## Development of predictive models for the three outcomes of interest

Table 4 highlights three models developed in this study, one for each outcome. Initially candidate predictors were selected from variables readily available within the inpatient clinical journey (including hospital discharge), as this would be most useful in clinical practice. Unfortunately, we were unable to isolate a model with acceptable levels of discrimination and calibration using these variables alone. In a second approach models were developed

using variables from the post discharge follow-up period (table 4). Three models with acceptable levels of calibration and discrimination were identified. All models require measures collected at hospital discharge, 1 month, and 3 months after discharge to predict the outcomes of interest.

**Model 1:** Poor physical function at 6 months after hospital discharge was predicted using the following: the combination of the measures of physical function both prior to injury and at 3 months after injury, a measurement of pain levels at 1 month after discharge and a measurement of the impact of pain on the individuals walking ability at hospital discharge. **Model 2:** It was possible to predict chronic pain at 6 months after hospital discharge using the following variables: Pre-injury regular use of analgesic agents, measures of physical function before the injury and at 3 months after discharge and a

**Table 4** Predictive models for three outcomes of interest

| | Coefficient | SE | OR | P value | 95% CI |
|---|---|---|---|---|---|
| **Model 1: poor physical function at 6 months after hospital discharge** | | | | | |
| Pre-injury PCS (range: 0–100) (one step increment increase) | −0.093 | 0.019 | 0.911 | <0.001 | 0.878 to 0.945 |
| Current pain at 1 month after injury (range 0–10) (one step increment increase) | 0.224 | 0.078 | 1.252 | 0.004 | 1.075 to 1.457 |
| PCS at 3 months (range: 0–100) (one step increment increase) | −0.040 | 0.014 | 0.960 | 0.005 | 0.934 to 0.988 |
| Impact of pain on walking at discharge (range: 0–10) (one step increment increase) | 0.139 | 0.057 | 1.149 | 0.015 | 1.027 to 1.285 |
| *Constant* | *3.745* | *1.132* | *–* | *0.001* | *–* |
| **Model 2: chronic pain at 6 months after hospital discharge** | | | | | |
| Regular analgesia use before injury | 1.813 | 0.533 | 6.130 | 0.001 | 2.158 to 17.414 |
| Pre-injury PCS (range: 0–100) (one step increment increase) | −0.039 | 0.019 | 0.962 | 0.044 | 0.926 to 0.999 |
| Pain score at 1 month after discharge (range: 0–10) (one step increment increase) | 0.270 | 0.089 | 1.310 | 0.003 | 1.099 to 1.560 |
| PCS at 3 months (range: 0–100) (one step increment increase) | −0.051 | 0.015 | 0.951 | 0.001 | 0.922 to 0.980 |
| Current pain score 3 months after discharge (range: 0–10) (one step increment increase) | 0.219 | 0.084 | 1.245 | 0.009 | 1.057 to 1.467 |
| *Constant* | *1.361* | *1.098* | *–* | *0.215* | *–* |
| **Model 3: likely neuropathic component to pain at 6 months after hospital discharge** | | | | | |
| Commenced on pregabalin during hospital admission | 2.340 | 1.156 | 10.377 | 0.043 | 1.076 to 100.113 |
| MCS at 1 month after discharge (range: 0–100) (one step increment increase) | −0.058 | 0.026 | 0.944 | 0.026 | 0.897 to 0.993 |
| Pain detect score at 3 months (range: −1 to 38) (one step increment increase) | 0.174 | 0.043 | 1.190 | <0.001 | 1.094 to 1.295 |
| Impact of pain on mood at 3 months after discharge (range 0–10) (one step increment increase) | 0.192 | 0.095 | 1.212 | 0.043 | 1.212 to 1.006 |
| *Constant* | *−1.421* | *1.188* | *–* | *0.232* | *–* |

MCS, Mental Component Score; PCS, Physical Component Score.

measurement of pain severity at 1 month and 3 months after discharge. **Model 3:** Neuropathic pain was predicted using the following variables: starting pregabalin during the acute hospital admission, the MCS at 1 month and the impact of pain on mood at 3 months after discharge, and the outcome of taking the PDQ at 3 months after discharge.

The ROC diagrams for each model are presented in figure 1. Figure 2 presents the graphitised calibration plots whereby plot points present risk of outcome in tenths of patients with similar predicted probabilities. The line highlights the relationship between the observed frequency and the predicted probability of developing a poor outcome. **Model 1** for predicting poor physical functioning at 6 months after discharge had a c-statistic (area under the receiver operator curve) of 0.846 (SE: 0.028; p<0.001; 95% CI 0.790 to 0.901) with a Hosmer and Lemeshow $\chi^2$ value of 5.570 (p=0.695) with 8 df, demonstrating acceptable goodness-of-fit. This resulted in a model with a sensitivity of 79% and a specificity of 81.2%. **Model 2** for predicting chronic pain at 6 months after discharge

had a c-statistic of 0.878 (SE: 0.023; p<0.001; 95% CI 0.833 to 0.923) with a Hosmer and Lemeshow $\chi^2$ value of 6.778 (p=0.561) with 8 df, also demonstrating acceptable goodness-of-fit. This resulted in a model with a sensitivity of 63.6% and a specificity of 86.5%. Finally, **Model 3** for predicting pain with a neuropathic component had a c-statistic of 0.910 (SE: 0.027;<0.001; 95% CI 0.857 to 0.964) with a Hosmer and Lemeshow $\chi^2$ value of 2.221 (p=0.912) with 8 df, demonstrating acceptable goodness-of-fit. This resulted in a model with a sensitivity of 61.3% and a specificity of 92.7%. The overall accuracy of models 1–3 were 80.6%, 78.1% and 85.8%, respectively. The calibration plots for each model (figure 2) demonstrated acceptable levels of calibration as many of the plots were close to the 45 degree line.

## DISCUSSION

This study investigated changes in self-reported HRQoL and pain outcomes over a 6-month period from hospital discharges after BTI. The study has provided new insights

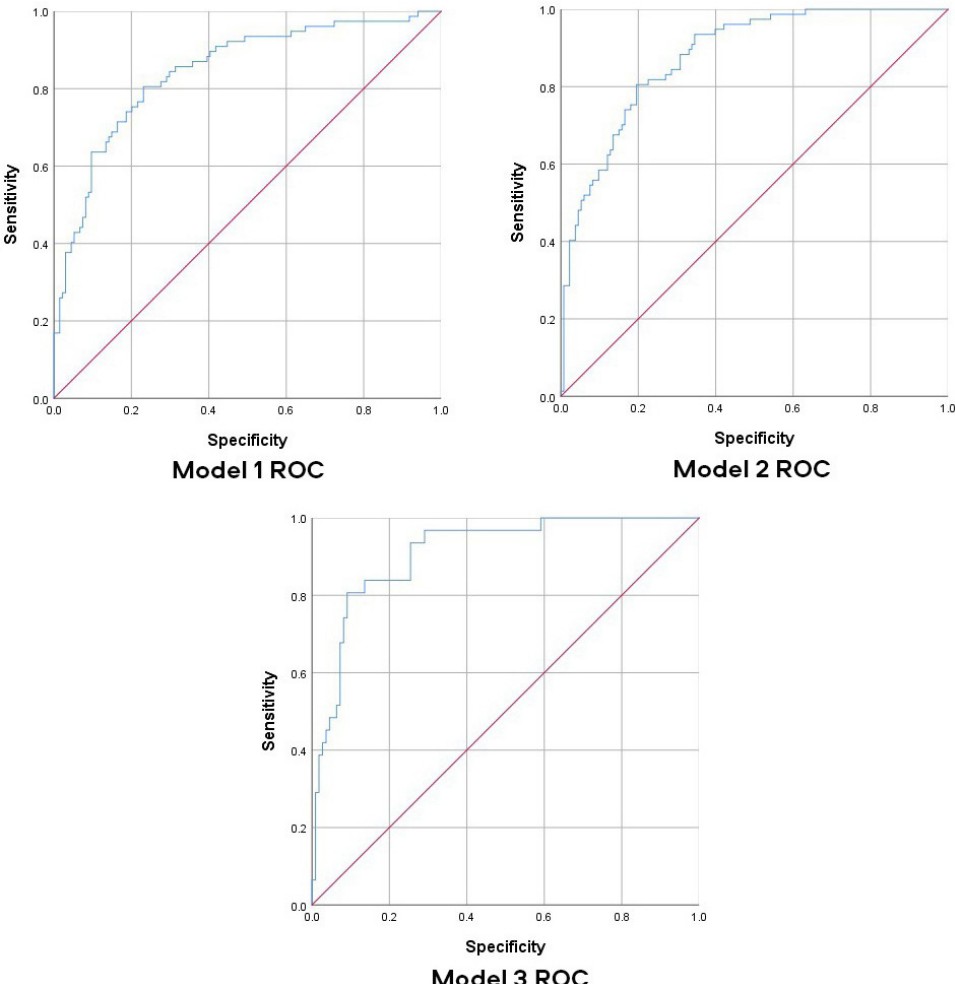

**Figure 1** Receiver operator curve (ROC) for predictive models for three outcomes of interest.

into the recovery of this patient group within the UK. This study has identified the high levels of pain and poor physical function experienced after BTI in the UK and measured the presence of neuropathic pain components within this population. Although it was not possible to construct a predictive model using only clinically available baseline and discharge level variables, the process of predictive prognostic modelling has identified factors that predict outcomes. This suggests that it is possible to develop a risk stratification score, but acknowledges that there is limited clinical utility in these current models as the data required to achieve accurate predictions of outcomes is not routinely available in clinical practice. Further research is needed to understand how clinically useful prediction models can be developed for this population. There is potential, with refinement, to make the current prediction models more clinically useful, and there are ways that this score could have clinical utility in identifying patients at risk of poor functional outcomes or chronic/neuropathic pain, subject to external validation in a future study. In respect to the models presented in this study, it is important to remember that good model performance in a development sample does

not necessarily mean good predictive abilities in a new external validation data set.

The limitations of using generic patient-reported outcome measures (PROMs) to measure HRQoL have been reported previously.[7 35] Although validated within the major trauma population, it remains unclear how specific and comprehensive generic outcome measures (eg, SF-12 and EQ-5D-5L) are in groups like the BTI population. Although a trauma PROM has previously been developed,[36] there is a need for injury specific measures, particularly where unique outcomes have been identified.[7 37] Physical functioning was measured at a global level in this study using the SF-12 PCS and using the individual levels of the EQ-5D-5L questionnaire. Both measures highlighted the extent of the altered physical function experienced by this population, which had not returned to their perceived baseline level of function at 6 months. In previous non-UK research, sequela related to BTI has been measured up to 2 years after injury highlighting the long-term impact of BTI.[38] Due to funding, it was not possible to extend the follow-up in the current study, but the 6-month time point was clinically ideal for the prognostic modelling process, particularly as Marasco

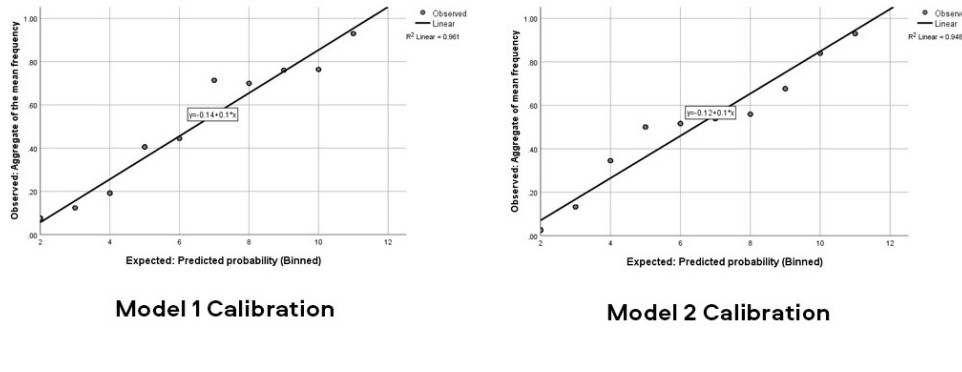

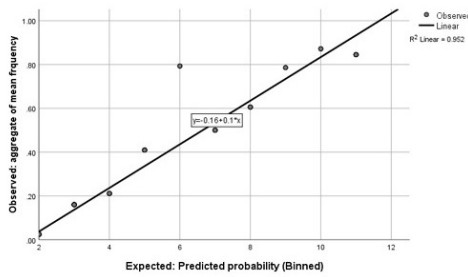

**Model 3 Calibration**

**Figure 2** Calibration plot for model using expected and observed probabilities of developing poor outcomes.

*et al* highlighted that patients reporting poor physical function at 6 months after injury reported poor physical function at 1 year and greater odds of reporting poor physical function at 2 years after injury.[38]

One key factor that influences HRQoL after BTI is pain, and the efficacy of pain management.[6] Measures of pain severity within this study highlighted 43.1% (n=91) of participants reported pain that was moderate, severe or extreme at 6 months after hospital discharge with a mean PSS of 2.7 (SD 2.7). For these participants, the mean analgesic effectiveness was 49.2 (SD 33.4) demonstrating how a chronic pain state existed where analgesic agents were not effective in managing pain. The variables in model 2 demonstrate the complexity in chronic pain management particularly in identifying those at risk through the predictive modelling process. Within this model it was not possible to predict the likelihood of whether a participant would experience chronic pain at 6 months after discharge using clinically available variables only. The final model included variables from pre-injury estimates through to 3 months after discharge which highlighted the need to alter our current model for patient follow-up to enable measurements at 1 and 3 months after discharge. Unsurprisingly, participants using regular analgesic agents prior to their injury had an OR>6 highlighting the need to identify patients with pre-existing chronic pain states during hospital admission as these patients are most likely to have ongoing injury-related pain requirements at 6 months after hospital discharge.

There were greater odds of reporting neuropathic pain at 6 months if the participant had been initiated on a neuropathic pain agent (eg, pregabalin) during the acute hospital admission. This suggests that clinical teams and specialist pain teams are accurately identifying patients at risk of neuropathic pain and commencing treatment. This unfortunately also highlights that these patients are still progressing to develop symptoms of neuropathic pain at 6 months after discharge which suggests that greater input from specialist pain services after hospital discharge may be required to manage this pain trajectory. The model for neuropathic pain highlights an association with the MCS of the SF-12 at 1 month and the effect of pain on mood at 3 months after hospital discharge. The relationship between neuropathic pain and mental health has previously been identified in the non-trauma literature.[39] Although there was a less substantial drop in SF-12 MCS compared with the PCS and this returned to the pre-injury baseline within the 6-month follow-up period, this shows the importance of measuring mental well-being as both a part of HRQoL and as an important predictor of neuropathic pain.

This study has developed a model for each of the three outcomes of interest: physical function, chronic pain and neuropathic pain at 6 months after hospital discharge with BTI. The next phase of the modelling process will be to conduct a further study to validate the predictive ability of the models within an external sample of participants with BTI. This study will further explore the follow-up needs of patients with BTI and look at effective ways of collecting these data needed to predict 6-month outcomes. It will also be of interest to investigate these outcomes beyond the 6 months follow-up period used in this study. In the

meantime, these predictive variables have clinical utility for clinicians reviewing patients during the follow-up period in primary and secondary care settings and these variables may be useful in highlighting patients at risk who need referral to specialist services (eg, physiotherapy or specialist pain services).

### Strengths and limitations of this study

Although we recruited a moderate sized sample using the exploratory approach to this study, we did not achieve the aim of recruiting 100 events relating to our three outcomes of interest. Care was taken to ensure that that the number of variables inputted into each model were justified by the number participants reporting the outcomes of interest. There are reports within the literature that the arbitrary rule of 10 cases per variable does not necessarily negatively impact on the model fit.[40] During subsequent validation of the models, care will be taken to assess for overfitting of the models but from the model diagnostics presented here, there is no suggestion of overfitting within the current sample.[32] It is important to remember that the final models are to some degree variable selection method dependent. In an ideal world, it would have been beneficial to use more than one method of selection but in the case of this study, it was not possible to undertake backward selection because of concerns of overfitting the model.

Overall, the combined response rate of 72% at all time points is comparable to equivalent studies undertaken previously in this area of trauma research.[6] Despite this, the response rate was a limitation as the key outcomes were measured at the 6-month time point resulting in only 63% (n=211) of the overall recruited sample being included in the modelling process. For further studies investigating outcomes within a longitudinal follow-up process, it will be important to use these data to consider potential sample sizes required to adequately power these studies.

Although multiple imputation was used in the management of missingness for independent variables, it is acknowledged that the case wise deletion approach used in the management of missingness in the outcome data are a limitation in this study. In the presence of MAR data, this approach results in large proportions of data being discarded, which results in the introduction of potential bias, and reduced precision and power. For this reason, and despite there being no clinically significant differences between the characteristics of responders and non-responders, the findings of this study should be interpreted cautiously. Despite all efforts to maximise response rates at all time points, future longitudinal studies in the BTI population will need to consider other ways of accessing patients to facilitate complete follow-up data collection in a geographically diverse population.

As subjective outcome measures have been used throughout this study, it is important to acknowledge the risk of bias which is inherent within these measures, particularly in relation to participants selecting the extremes of a rating scale.[41] Care has been taken within the study to control this bias through the instructions given to patients prior to completing each questionnaire and investigation of outliers within the sample during data analysis.[42] In future research, it would be interesting to compare the predictive ability of both subjective outcomes (ie, SF-12 PCS for physical function) with an objective measure (grip strength testing as an estimate of physical functional ability for example) and compare the difference in the predictive functions of these outcomes.[43]

Finally, although most of the study was undertaken before the COVID-19 pandemic began in the UK, there will be seen and unseen impacts of the COVID-19 pandemic on this study.[44] The final three sites closed to recruitment 2–4 weeks early causing them to not meet recruitment targets. Furthermore, follow-up continued throughout the first wave of the pandemic in the UK. Although we were not able to collect data relating COVID-19 infection among our sample, we were not informed that any participants had been infected with the virus. Despite this, there is potential bias relating to lockdown, social isolation that remains unmeasured within our sample which may have impacted on the self-reported outcome measures reported in this study.[45]

## CONCLUSIONS

It remains vital that the interdisciplinary team do not underestimate the impact of the long-term outcomes associated with BTI, particularly relating to physical functioning, chronic pain and neuropathic pain. Although predictive models were successfully developed, this study has demonstrated the complexity and challenge of developing models which are clinically useful as many of the variables included in these final models are not routinely available in practice. Despite this, the study has shown how predictive models can be developed for outcomes and highlight how PROMs can be used to develop prognostic models particularly in longitudinal processes where objective clinical data are not routinely collected. Ultimately, these models add to our understanding of which factors influence the development of sequela after hospital discharge with BTI and after external validation, will have clinical utility in the risk stratification of high-risk patient groups.

**Author affiliations**
[1]Florence Nightingale Faculty of Nursing, Midwifery and Palliative Care, King's College London, London, UK
[2]Emergency Department, King's College Hospital, London, UK
[3]Welsh Institute of Biomedical and Emergency Medicine Research, Swansea Bay University Health Board, Port Talbot, UK
[4]Emergency Department, Buckingham Healthcare NHS Trust, Amersham, UK
[5]Emergency Department, North Bristol NHS Trust, Westbury on Trym, UK
[6]Emergency Department, Surrey and Sussex Healthcare NHS Trust, Redhill, UK
[7]Department of Anesthetics, Aneurin Bevan Health Board, Newport, UK
[8]Critical Care, King's College Hospital NHS Foundation Trust, London, UK
[9]Emergency Department, Barnsley Hospital NHS Foundation Trust, Barnsley, UK
[10]Emergency Department, St George's Healthcare NHS Trust, London, UK
[11]General Surgery, Aneurin Bevan Health Board, Newport, UK

[12] Emergency Department, Barts Health NHS Trust, London, UK
[13] Emergency Department, University Hospitals of Derby and Burton NHS Foundation Trust, Derby, UK
[14] Emergency Department, King's College Hospital NHS Foundation Trust, London, UK
[15] Emergency Department, Imperial College Healthcare NHS Trust, London, UK

**Acknowledgements** The research team would like to acknowledge the study participants who gave up their time to participate in this interview study and the clinical research staff at each site who contributed to the participants recruitment and initial data collection for the Rib Injury Outcomes Study.

**Contributors** EB, GL, AX, CN and PH conceived and designed the study and developed the study protocol. AB, CB, EC, CD, JF, RJ, LP, AR, IS, AT, MT and LY were local primary investigators and coordinated and contributed significantly to the recruitment and data collection processes. Data analysis was planned with EB, GL, TM and CB. Data analysis was undertaken by EB and reviewed by TM. EB developed the first draft of the manuscript and EC, CB, AX, CN, PH and GL made subsequent revisions. All authors commented on and approved the manuscript prior to submission for publication. EB takes overall responsibility for the manuscript.

**Funding** EB is funded by a National Institute for Health Research (NIHR), Clinical Doctoral Research Fellowship for this research project (ICA-CDRF-2016-02-006). The publication presents independent research funded by the NIHR. The views expressed are those of the authors and not necessarily those of the National Health Service, the NIHR or the Department of Health and Social Care.

**Competing interests** None declared.

**Patient consent for publication** Not required.

**Ethics approval** Ethical approval was granted by the 'Hampshire A' South Central Research Ethics Committee in June 2018 (ref: 18/SC/0230).

**Provenance and peer review** Not commissioned; externally peer reviewed.

**Data availability statement** All data relevant to the study are included in the article or uploaded as supplementary information. The authors confirm that the data supporting the findings of this study are available within the article and its supplementary materials.

**ORCID iDs**
Edward Baker http://orcid.org/0000-0002-2633-0871
Imogen Skene http://orcid.org/0000-0001-5215-2899
Andrew Tabner http://orcid.org/0000-0003-4191-9024
Andreas Xyrichis http://orcid.org/0000-0002-2359-4337

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
