## [Reviewer comments · BMJ Open]

ARTICLE DETAILS

TITLE (PROVISIONAL)	A prospective observational study to examine health related quality of life and develop models to predict long-term patient reported outcomes six-months after hospital discharge with blunt thoracic injuries.
AUTHORS	Baker, Edward; Battle, Ceri; Banjeri, Abhishek; Carlton, Edward; Dixon, Christine; Ferry, Jennifer; Hopkins, Philip; Jones, Robert; Murrells, Trevor; Norton, Christine; Patient, Lee; Rasheed, Ashraf; Skene, Imogen; Tabner, Andrew; Tunnicliff, Malcolm; Young, Louise; Xyrichis, Andreas; Lee, Gerry

VERSION 1 – REVIEW

REVIEWER	Carrie, Cédric Centre Hospitalier Universitaire de Bordeaux
REVIEW RETURNED	12-Feb-2021

GENERAL COMMENTS	I have read with a great interest the paper untitled: "A prospective observational study to examine health related quality of life and develop models to predict long-term patient reported outcomes six-months after hospital discharge with blunt thoracic injuries ». In this prospective multicenter observational study, the authors aimed to (1) examine the 6-month outcomes and HRQoL in blunt chest trauma patients, (2) develop predictive models for poor physical functioning, chronic pain and neuropathic pain over 6 months after hospital discharge. Over the study period, 377 patients were included but only 211 (63%) completed the 6-month follow-up for the main study outcomes. Authors reported high rates of poor physical functioning (37%), chronic pain (37%) or neuropathic pain (22%). The predictive models were developed using post-discharge follow up. Overall, it is a generally well-conducted research and a well-written document, although I have some comments for the authors' consideration before publication. Major comments: 1) Statistical analysis of missing data: The large number of patients with missing data (N = 126, 37%) corresponds to a very significant bias which can greatly modify the results, especially for the modelling process. In this context, I am not comfortable with the statistical analysis of missing data. I strongly recommend a specialized statistician opinion. 2) Clinical interpretation of the results: As noticed in the "introduction" section, follow-up after discharge is very limited, hence predictive models would be most useful if the covariates were available before hospital discharge. Except for model 1, more than 50% of the variables used in their predictive model
--

	need a 3-month follow-up, which is in contradiction with the background of the present study. I would appreciate to know the position of the authors on the clinical utility of such predictive models requiring a long-term follow-up...Please develop this point in the discussion section. Minor comments are indicated by line number in order that they appear in the manuscript.  - Table 1. Define "complications"; please correct the IQR of hospital LOS ; please add some other valuable covariates (seen in supplementary data) to better describe the management of the population (use of locoregional analgesia, need for chest tube insertion, surgical fixation of rib fracture...). - Table 2.  o Please remind the sample size of patients analyzed at the different times (pre-injury, discharge, 1-month...). o Please define PDQ in the methods section (abbreviation not mentioned P.9 ; L.237). o Remove the (*) o Most of the results in Table 2 are presented as median [25 – 75% IQ]. To our point of view, a Wilcoxon test for paired values should be more appropriate and should be noticed in the "statistical analysis" section. - Table 3. Define the bold font and/or * = $p < 0.05$ between pre-injury and 6-months - Supplementary Data. The results suggest a significant association between clinical long-term outcome and several covariates of interest (older age, previous comorbidities, previous analgesic use, surgical rib fixation, need for mechanical ventilation and/or ICU admission...). Those covariates should appear in Table 1 rather than in supplementary data S1.
--	---

REVIEWER	Schroepel, Thomas University of Colorado Colorado Springs
REVIEW RETURNED	26-Feb-2021

GENERAL COMMENTS	Well written very interesting manuscript. I found that I got lost in the statistics and models. I think the limitations section needs to be expanded to comment more on the COVID lockdown and potential implications on the surveys. I think we are all going to find out that the impacts of the lockdowns are more profound than any of us currently realize. I think the discussion on the use of the predictive models needs to be expanded. The development of the models and the process is very interesting, but how in practice are we going to use them. Overall, the manuscript is too long and could benefit from being more concise. Would recommend review by statistician to review model development and imputation technique.
--

REVIEWER	Pontiroli, Antonio University of Milan, medicina
REVIEW RETURNED	21-Mar-2021

GENERAL COMMENTS	the authors should state that patients not included in ths study were not different from patients not included
--

REVIEWER	Brown, Siobhan University of Washington, Department of Biostatistics
REVIEW RETURNED	02-Apr-2021

GENERAL COMMENTS

I find the missing outcome data insufficiently handled. Though this decision was made 'a priori,' it was a poor one that makes the study results basically uninterpretable. Further, the longitudinal aspect of the observations was not fully addressed.

Study methods: Were any additional attempts to collect missing outcomes? Perhaps follow-up phone calls to those who did not complete the survey? Was contact information collected at hospital discharge? Convince the reader that you made reasonable attempts to gather complete data.

Statistical methods:
The Chi-squared test can fail when the expected cell count is less than five, rather than the observed.

The t-test is quite robust to departures from the assumption of normality. With the sample sizes you have here, the Mann-Whitney test is not necessary.

p. 13, lines 274-275: rather than describing the SPSS programming "In this process, SPSS...", it would be preferable to reference the underlying statistical methodology (Rubin's rule, I presume).

The positive predictive value and negative predictive values are highly dependent on the population prevalence. With the amount of missing outcomes/complete case analysis, they are not meaningful and may be misleading.

You should provide a justification for using a binary rather than continuous outcomes in the predictive models. Readers may be interested in the details of the distribution of the outcomes. A histogram or similar would give us a good sense of whether the distribution was bi-modal or heavily skewed, for instance.

Please make it clear which subjects are included in each analysis.

The large amount of missing outcome data, while not surprising in long-term follow-up of trauma patients, must be directly addressed. You cannot make any claims about the longitudinal progression of outcomes from the current analysis. It appears to me that number of subjects changes across the different time points! I suggest looking at the baseline characteristics of those with complete outcome data vs. those with some missing outcome measures. If they are similar, you might perhaps be able to argue for inference from only the complete cases. If, as I suspect, they are not, you'll need to use multiple imputation or some sort of weighting to account for the potential bias introduced by selective dropout.

When you analyze changes over time, are you summarizing changes for each individual, or the differences in the group means at the different measurement points? Are the same subjects included for all analyses?

Discussion:
I would mention again that you were not able to develop a predictive model using only baseline/discharge variables. To the sentence, "The process of predictive prognostic modelling has identified factors that predict outcomes, suggesting that it is

	possible to develop a risk stratification score,” perhaps append, “though not with variables available at baseline.” p. 25, line 535: I believe you mean “input” rather than “imputed.”
--	--

VERSION 1 – AUTHOR RESPONSE

	Reviewer comments	Author responses
Reviewer 1		
1.1	Statistical analysis of missing data: The large number of patients with missing data (N = 126, 37%) corresponds to a very significant bias which can greatly modify the results, especially for the modelling process. In this context, I am not comfortable with the statistical analysis of missing data. I strongly recommend a specialized statistician opinion.	Thank you for this comment. Mr Trevor Murrells is the study Statistician and co-author on the paper. He has been involved in the development and implementation of a statistical analysis plan and we have reviewed the statistical plan and statistics used in the manuscript as part of this this revision. Due to the limitation of the journal word count, we were unable to add further details about missingness and missing data analysis in the original manuscript. We recognise the importance of this and have subsequently included a section in the results about the missing data analysis. In this new section, the following paragraph was added to data already included in the manuscript: ‘A missing data analysis was undertaken as part of the analysis process to identify any differences in the characteristics of participants who did or did not respond to follow-up at six months after hospital discharge when outcome data were collected. Supplementary table S5 presents the results from this analysis. Although there was a statistically significant difference in participants age, incidence of cardiac disease and cancer, and MCS at the pre-injury, discharge and one-month post discharge timepoint, these were not deemed to be clinically significant differences. Furthermore, a logistic regression model was built using a stepwise forward approach using variables identified in the univariable analysis of missingness to identify predictors of responder status at six months after discharge. It was not possible to fit a model using the selected variables and no statistically significant predictors were identified. Since none were identified, it was assumed that the two groups

		were characteristically similar and therefore a sensitivity analysis, using weights calculated from the logistic regression model, was not undertaken. Based on Little and Rubin's taxonomy missingness was assumed to be Missing at Random (MAR) at the unit level and multiple imputation methods in SPSS were used to address item level missingness (Ayilara et al., 2019; Schlomer, Bauman, & Card, 2010).' A further supplementary table has been added to include data from the analysis of responders vs. non-responders.
1.2	Clinical interpretation of the results: As noticed in the "introduction" section, follow-up after discharge is very limited, hence predictive models would be most useful if the covariates were available before hospital discharge. Except for model 1, more than 50% of the variables used in their predictive model need a 3-month follow-up, which is in contradiction with the background of the present study. I would appreciate to know the position of the authors on the clinical utility of such predictive models requiring a long-term follow-up...Please develop this point in the discussion section.	The study aim was to develop models with high levels of clinical utility. The reality after analysis was that this was not possible, and the outputs have been acknowledged to be of limited clinical utility. A statement has been added to the discussion to this end alongside the need for further investigation to identify methods of achieving clinical utility from the models which are presented within the manuscript. The following has been added: 'It is acknowledged there is limited clinical utility in these current models as the data required to achieve accurate predictions of outcomes is not routinely available in clinical practice. Further research is needed to understand how clinically useful prediction models can be developed for this population. There is potential, with refinement, to make the current prediction models more clinically useful, and there are ways that this score could have clinical utility in identifying patients at risk of poor functional outcomes or chronic/neuropathic pain, subject to external validation in a future study.'
1.3	Table 1. Define "complications"; please correct the IQR of hospital LOS ; please add some other valuable covariates (seen in supplementary data) to better describe the management of the population (use of	 • 'Respiratory' has been added to complications in this table to provide a clearer meaning. • IQR has been converted to Mean with Standard Deviations (see comment 1.4 and 4.4).

	locoregional analgesia, need for chest tube insertion, surgical fixation of rib fracture...).	 • Further additional co-variates have been added to Table 1 from the Supplementary table.
1.4	Table 2.  • Please remind the sample size of patients analyzed at the different times (pre-injury, discharge, 1-month...). • Please define PDQ in the methods section (abbreviation not mentioned P.9 ; L.237). • Remove the (*) • Most of the results in Table 2 are presented as median [25 – 75% IQ]. To our point of view, a Wilcoxon test for paired values should be more appropriate and should be noticed in the “statistical analysis” section. 	 • Sample size at each time-point has been added into table 2. • The Abbreviation PDQ has been added alongside the definition of the Pain DETECT questionnaire in the methods section. • (*) have been removed from statistically significant p-values. • As per reviewer 4 comments, and after discussion with the study statistician, data is now presented as mean values with standard deviations and comparisons of independent means using student T-test.
1.5	Table 3. Define the bold font and/or * = p < 0.05 between pre-injury and 6-months	The bold font was used to differentiate data from the total sample from the comparative analysis in the same table. This has now been removed to save any further confusion.
1.6	Supplementary Data. The results suggest a significant association between clinical long-term outcome and several covariates of interest (older age, previous comorbidities, previous analgesic use, surgical rib fixation, need for mechanical ventilation and/or ICU admission...). Those covariates should appear in Table 1 rather than in supplementary data S1.	Further descriptive variables have now been added to the tables within the main manuscript.
Reviewer 2:		
2.1	Well written very interesting manuscript. I found that I got lost in the statistics and models.	Thank you for these supportive comments. We have undertaken a review of the statistical components of this manuscript and made

		revisions where required to ensure that this is easy to understand and comprehend.
2.2	I think the limitations section needs to be expanded to comment more on the COVID lockdown and potential implications on the surveys. I think we are all going to find out that the impacts of the lockdowns are more profound than any of us currently realize.	We completely agree with the feedback in relation to the potential impact of lockdown and social distancing on aspects of HRQoL within all parts of the population. This has been clearly stated in the original manuscript, but as this has not yet been realised, we feel that there is no need to further elaborate on this within the revised manuscript.
2.3	I think the discussion on the use of the predictive models needs to be expanded. The development of the models and the process is very interesting, but how in practice are we going to use them.	We agree also that it needs to be clearly stated that the research team acknowledge that the models produced within this study do not have the clinical utility that we hoped we would achieve. The following statement has been added to the discussion section of this manuscript: 'It is acknowledged there is limited clinical utility in these current models as the data required to achieve accurate predictions of outcomes is not routinely available in clinical practice. Further research is needed to understand how clinically useful prediction models can be developed for this population. There is potential, with refinement, to make the current prediction models more clinically useful, and there are ways that this score could have clinical utility in identifying patients at risk of poor functional outcomes or chronic/neuropathic pain, subject to external validation in a future study.'
2.4	Overall, the manuscript is too long and could benefit from being more concise.	The manuscript has been revised and where possible cuts have been made to ensure the revised manuscript is as concise as possible.
2.5	Would recommend review by statistician to review model development and imputation technique.	The study statistics have been reviewed by the study statistician, the model development processes, and imputation approach have been reviewed discussed and revised in the manuscript. The statistician is one of the authors of the paper and has been involved in all aspects of manuscript preparation.
Reviewer 3		

3.1	The authors should state that patients not included in this study were not different from patients not included	Missingness within the data set and a missing data analysis has been dealt with after comments from reviewer 1. Please see comment 1.1 in this document.
Reviewer 4		
4.1	I find the missing outcome data insufficiently handled. Though this decision was made 'a priori,' it was a poor one that makes the study results basically uninterpretable. Further, the longitudinal aspect of the observations was not fully addressed.	During data collection process, the potential need for longitudinal imputation for outcome variables was discussed with the study statistician. The decision was made to not use this method due to the high level of missingness at both 1-month and 3 months and the low number of participants who completed the follow-up data collection at all timepoints. Rather a case wise approach was used. An analysis of responders vs. non-responders was conducted and no clinically significant differences were identified between participants who responded and those did not. Furthermore, a logistic regression model was built to identify predictors of non-response and no predictors were identified. A clear statement of missingness and analysis of responders vs. non-responders has been included within the revised manuscript. This is included in this table with the comment 1.1. In relation to the longitudinal imputation of outcome variables, the following statement has been added to the methods section: 'Longitudinal imputation of outcome variables was considered but not conducted in this study due to the high level of missingness at both 1-month and 3 months and the low number of participants who completed the follow-up data collection at all timepoints. Rather a case wise approach was used.' The authors acknowledge the limitation of using a case wise deletion approach in relation to the outcome variables in the study, we would argue that the findings are no uninterpretable. We have subsequently added the follow statement within the limitations section of the revised manuscript to ensure

		that the findings from this study are interpreted with caution: ‘Although multiple imputation was used in the management of missingness for independent variables, it is acknowledged that the case-wise deletion approach used in the management of missingness in the outcome data is a limitation in this study. In the presence of MAR data, this approach results in large proportions of data being discarded, which results in the introduction of potential bias, and reduced precision and power. For this reason, and despite there being no clinically significant differences between the characteristics of responders and non-responders, the findings of this study should be interpreted cautiously. Despite all efforts to maximise response rates at all timepoints, future longitudinal studies in the BTI population will need to consider other ways of accessing patients to facilitate complete follow-up data collection in a geographically diverse population.’
4.2	Study methods: Were any additional attempts to collect missing outcomes? Perhaps follow-up phone calls to those who did not complete the survey? Was contact information collected at hospital discharge? Convince the reader that you made reasonable attempts to gather complete data.	Participants were contacted three times with access to the survey using a multimedia approach. The following statement has been added to the methods section to this effect: ‘During post discharge follow-up non-responding participants were contacted up to three times using a mixed multi-media approach including telephone, post, or email, to maximise participation and minimise the potential for missing data.’
4.3	Statistical methods: The Chi-squared test can fail when the expected cell count is less than five, rather than the observed.	This has been updated within the manuscript.
4.4	The t-test is quite robust to departures from the assumption of normality. With the sample sizes you have here, the Mann-Whitney test is not necessary.	Thank you for this advice. I have taken this onboard and calculated t-tests for the comparisons of continuous variables and have now presented all continuous variables as

		means with standard deviations within both the manuscript narrative and the tables. The statistical methods section of the manuscript has been updated to recognise this change in analysis method: ‘Continuous data were presented as means with standard deviations. The statistical tests applied for binary/categorical data were Chi-square test of association, except where the expected cell counts were ≤ 5 when the Fisher’s exact test was used. Comparisons continuous data from two independent groups was undertaken using the Student t-test.’
4.5	p. 13, lines 274-275: rather than describing the SPSS programming “In this process, SPSS...,” it would be preferable to reference the underlying statistical methodology (Rubin’s rule, I presume).	This sentence has now been removed as a missing data analysis section has been added to the results section of the manuscript. This describes the process of reaching the conclusion of MAR and introduced the Little and Rubin Taxonomy formally (see statement in comment 1.1).
4.6	The positive predictive value and negative predictive values are highly dependent on the population prevalence. With the amount of missing outcomes/complete case analysis, they are not meaningful and may be misleading.	Both PPV and NPV have been excluded from the revised manuscript.
4.7	You should provide a justification for using a binary rather than continuous outcomes in the predictive models. Readers may be interested in the details of the distribution of the outcomes. A histogram or similar would give us a good sense of whether the distribution was bi-modal or heavily skewed, for instance.	A justification has been added within the methodology section for using binary outcomes rather than continuous. This statement also acknowledges the limitations of creating binary outcomes from continuous variables. Unfortunately, due to the limitations of both word count and the restrictions on the number of tables and figures it will not be possible to add further information about the shape of distribution in this current manuscript.
4.8	Please make it clear which subjects are included in each analysis.	This has been clarified within the revised manuscript.

4.10	When you analyze changes over time, are you summarizing changes for each individual, or the differences in the group means at the different measurement points? Are the same subjects included for all analyses?	This has been clarified within the statistical analysis section of the methods. The following statement has been added: 'Longitudinal changes in continuous variables over the course of the post discharge follow-up period are reported at a group level. This uses the longitudinal change in the group mean from one timepoint to the next to measure change.'
4.11	Discussion: I would mention again that you were not able to develop a predictive model using only baseline/discharge variables. To the sentence, "The process of predictive prognostic modelling has identified factors that predict outcomes, suggesting that it is possible to develop a risk stratification score," perhaps append, "though not with variables available at baseline."	This has been clarified within the discussion section of the methods. The following statement has been added: 'Although it was not possible to construct a predictive model using only clinically available baseline and discharge level variables, the process of predictive prognostic modelling has identified factors that predict outcomes. This suggests that it is possible to develop a risk stratification score, but this involves using variables which are not routinely available in clinical practice currently. It is acknowledged there is limited clinical utility in these current models as the data required to achieve accurate predictions of outcomes is not routinely available in clinical practice. Further research is needed to understand how clinically useful prediction models can be developed for this population.'
4.12	p. 25, line 535: I believe you mean "input" rather than "imputed."	This has been corrected in the main manuscript.

VERSION 2 – REVIEW

REVIEWER	Carrie, Cédric Centre Hospitalier Universitaire de Bordeaux
REVIEW RETURNED	03-May-2021
GENERAL COMMENTS	We thank the authors for taking into account previous comment. Before publication, two minors comments should be made : 1) There is a typo in the second paragraph of the statistical analysis (participants) 2) The conclusion is very optimistic about the interest of predictive models ; on th other hand, the authors acknowledged there is

	limited clinical utility in these current models as the data required to achieve accurate predictions of outcomes is not routinely available in clinical practice. The conclusion should mention this point.
--	--

REVIEWER	Schroepfel, Thomas University of Colorado Colorado Springs
REVIEW RETURNED	18-May-2021

GENERAL COMMENTS	I appreciate the response to the queries from all reviewers by the authors. I still feel that this study is biased from the missing data and that is not overcome by the analysis of the missing data and the imputation methods. I also question the clinical utility of these models. I think those issues are fatal flaws in this manuscript.
--

REVIEWER	Brown, Siobhan University of Washington, Department of Biostatistics
REVIEW RETURNED	20-May-2021

GENERAL COMMENTS	Though I still have reservations about the handling of missing data, I believe the authors have fully acknowledged the limitations of their analysis and the paper is now acceptable for publication.
---

VERSION 2 – AUTHOR RESPONSE

Reviewer Comments:	Author Response:
Reviewer 1:	
We thank the authors for taking into account previous comment. Before publication, two minors comments should be made:  1. There is a typo in the second paragraph of the statistical analysis (participants) 2. The conclusion is very optimistic about the interest of predictive models; on the other hand, the authors acknowledged there is limited clinical utility in these current models as the data required to achieve accurate predictions of outcomes is not routinely available in clinical practice. The conclusion should mention this point. 	The authors would like to thank all reviewers for taking the time to review this manuscript and comment on the previously submitted revisions. This typo has now been corrected within the revised manuscript. The conclusion section of the manuscript has been revised and a sentence has been added to highlight the limited clinical utility in the current model as it stands. 'Although predictive models were successfully developed, this study has demonstrated the complexity and challenge of developing models which are clinically useful as many of the variables included in these final models are not routinely available in practice'

Reviewer 2:

I appreciate the response to the queries from all reviewers by the authors. I still feel that this study is biased from the missing data and that is not overcome by the analysis of the missing data and the imputation methods. I also question the clinical utility of these models. I think those issues are fatal flaws in this manuscript.

Thank you for your further comment. In the revised manuscript, I have fully acknowledged the issue of missingness within the limitations section of the manuscript and have advised readers to interpret the study results with caution due to these limitations. As a group the authors continue to feel that the manuscript adds to the current evidence base and will be of interest to the readership of the journal and the wider trauma community. We believe that these models are a useful starting point for further development, refinement and validation processes and this manuscript adds an important reference point in the process of developing prognostic models for long-term outcomes after blunt thoracic injury. Furthermore, revisions have been made in the manuscript to ensure the limited clinical utility of the current models has been clearly stated.

Reviewer 4:

Though I still have reservations about the handling of missing data, I believe the authors have fully acknowledged the limitations of their analysis and the paper is now acceptable for publication.

We have discussed the statistical methods for the management of missing data with the study statistician (who is an author on this paper) and we believe that we have been fully transparent and acknowledged the potential limitations within the method relating to management of missingness within the data-set. I would like to thank you for your comments throughout the review process as these have been very helpful in strengthening of the manuscript in preparation for publication in this journal.